# Refined Analysis of the Transient Temperature Effect during the Closing Process of a Cross-Sea Bridge

**Zuolong Luo** [1], **Yuan Li** [2,*], **Jiaqing Wang** [3] **and Fenghui Dong** [3]

1 School of Electric Power, Civil Engineering and Architecture, Shanxi University, Taiyuan 030006, China; luozuolong@sxu.edu.cn
2 School of Highway, Chang'an University, Xi'an 710064, China
3 College of Civil Engineering, Nanjing Forestry University, Nanjing 210037, China; jiaqingw@njfu.edu.cn (J.W.); nldfh@njfu.edu.cn (F.D.)
* Correspondence: liyuan@chd.edu.cn

**Abstract:** In order to study the transient temperature effect during the construction of a cross-sea bridge off the coast, based on the Hong Kong-Zhuhai-Macao Bridge-Pipe Bridge Crossing Cliff 13-1 Gas Field, a refined analysis was conducted and the prediction of transient temperature gradient and structural response was carried out under the conditions of strong solar radiation and atmospheric convection using the method of combining theoretical research and numerical simulation. Firstly, the partial differential equation of uniform heat flux density on the outer surface of the main girder under the action of solar radiation and atmospheric convection was established. The equation was realized by calculating the solar radiation intensity and the comprehensive heat transfer coefficient, as well as fitting the atmospheric temperature on the outer surface of the main girder, and the equivalent comprehensive temperature at any time on the main girder was obtained. Secondly, a numerical analysis model of the heat conduction of the main girder section was established, and the equivalent comprehensive temperature was input into the numerical model as the temperature field boundary to solve the transient temperature gradient of the section, and the result was verified in comparison with the measured data. Finally, the transient temperature gradient was applied to the girder, and the temperature effect of the main girder during the closing process was also calculated. Construction control measures were also discussed. The research results show that the predicted value of the transient temperature gradient is consistent with the measured value (the maximum deviation is less than 2 °C), and the predicted value is slightly larger than the measured value, which makes the structure safer. During the closing process, the temperature gradient of the main girder has obvious non-linear characteristics: the temperature gradient is relatively high within 0.4 m of the top surface of the roof while tending to zero outside 0.4 m. The best closing time for the main girder is from 21:00 in the evening of the closing day to 6:00 a.m. the next day. For the small angles at both ends of the closure segment during the best closing time, temporary adjustment jacks and temporary counterweights can be adopted to eliminate the small angles at both ends of the closure segment in order to facilitate the welding construction and meet the smoothness requirements of bridge alignment.

**Keywords:** bridge engineering; transient temperature effect; theoretical research; numerical simulation; cross-sea bridge; refined analysis

## 1. Introduction

During the construction of sea crossing bridges, due to the influence of solar radiation, atmospheric convection, annual temperature change, sudden cooling, and other factors, uneven temperature fields will be generated inside the structure, and the resulting temperature stress and deformation will adversely affect the structure [1–6]. The temperature effect of bridge structures can generally be summarized into three types: annual temperature

change, sunshine temperature, and sudden cooling [7–14]. For cross-sea bridges that have been exposed to the marine environment for a long time, the sunlight temperature effect is particularly significant. Under extreme conditions, the temperature effect caused by strong sunlight radiation and atmospheric convection even exceeds the dead load and live load effects and becomes the first control factor in bridge design and construction. It seriously threatens the smooth construction and normal operation and maintenance of cross-sea bridges.

Many scholars have carried out extensive research on the temperature effect of bridge structures, mainly using the measured data of the temperature field combined with the finite element numerical simulation to study the temperature field distribution and temperature effect of bridges of various materials and structural forms under external meteorological conditions, and have obtained some valuable conclusions. Abid et al. [15] conducted experimental research on the temperature field of concrete beams with box sections, analyzed the impact of environmental temperature fluctuations on the temperature field distribution of box sections through measured temperature field data, and proposed a formula for calculating the temperature gradient of box sections under the action of environmental temperature. Yarnold et al. [16] used the health monitoring system to monitor the long-term temperature effect of long-span bridges, studied the generation mechanism of the structural temperature effect through the measured temperature field data, and obtained the temperature gradient distribution mode of key sections of the structure; Maguire et al. [17] fitted the temperature gradient of the main beam section of Varina Enon Bridge under the external meteorological conditions based on the measured data of long-term temperature field, and evaluated the thermal stress damage of the structure under the effect of temperature gradient; Gao Jing et al. [18], based on the measured data of temperature field, fitted the cumulative distribution function of temperature difference between different measuring points on the same section of flat steel box girder, and thus established the calculation model of temperature gradient of steel box girder; Ding Youliang et al. [19] studied the distribution characteristics of horizontal temperature difference and vertical temperature difference of flat steel box girder based on long-term temperature monitoring results, established a two-dimensional temperature gradient model of steel box girder, and discussed the similarities and differences of temperature gradient models of suspension bridge and cable-stayed bridge; Lin Caikui et al. [20] established positive and negative temperature difference functions along the height of a continuous rigid frame bridge with box section in Guangdong Province through temperature monitoring and temperature difference data analysis and compared them with the temperature gradient function specified in the specifications; Liu Jiang et al. [21] derived the representative value of the temperature difference of the box section by using the generalized extreme value distribution based on the measured results of the temperature field of the concrete box girder, calculated the temperature difference of the box section in 34 major cities in China, and clarified the regional differences of the temperature difference values.

The existing research mainly focuses on the temperature field distribution mode of inland bridges under conventional meteorological conditions. However, there are not enough scientific studies and analyses focusing on temperature field distribution and transient temperature effects of offshore bridges under strong solar radiation and atmospheric convection environments. Additionally, most of the existing studies are based on measured temperature data, which cannot accurately predict the temperature effect of the sea crossing bridge at any time in the absence of on-site measured data. For continuous steel box girder bridges constructed by the large segment assembly method, the temperature effect will have a great impact on the length of the closure segment, the smoothness of the closure joint, and the finished bridge alignment. Predicting the temperature effect of the structure at any time in the closure process in advance and selecting the appropriate closure time to ensure a smooth closure process is a technical problem faced in the current construction process of sea crossing bridges [22]. Therefore, in the absence of measured temperature data, refined analysis and prediction of the transient

temperature gradient model of cross-sea bridges under complex marine meteorological conditions, such as strong sunlight radiation and atmospheric convection, are essential to fully reflect the distribution characteristics of the overall transient temperature field of the structure and achieve the accurate calculation of the transient temperature effect during the key construction stages of cross-sea bridges. This is an important subject that requires corresponding construction control measures to ensure the smooth construction of cross-sea bridges and needs further study.

Taking the Hong Kong Zhuhai Macao Bridge Crossing Ya 13-1 Gas Field Pipeline Bridge as the engineering background, this paper addresses the issue of refined analysis of temperature field distribution and the transient temperature effect of offshore bridges under strong solar radiation and atmospheric convection environments, using the method of combining theoretical research and numerical simulation. The new approach also allows for a more efficient and effective solution to the complex problem of predicting the temperature effect of the sea crossing bridge at any time in the absence of on-site measured data. Additionally, the proposed method provides important guidance for the optimal timing of closure and the selection of construction control measures during the closure process. Overall, this study is important as it offers a new approach to accurately analyzing and predicting the temperature gradient effect during the closure process of a cross-sea bridge, proposes reasonable construction control measures, and ensures the smooth construction of the cross-sea bridge.

## 2. Project Introduction

### 2.1. Project Overview

The Hong Kong Zhuhai Macao Bridge spans the Lingdingyang Sea area, connecting Hong Kong in the east and Zhuhai and Macao in the west. The main project is about 29.6 km long, which is the longest cross-sea bridge project in the world. The pipeline bridge across Ya 13-1 Gas Field belongs to the non-navigable span bridge of section DB01 of the Hong Kong Zhuhai Macao Bridge. The span combination is (110 + 150 + 110) m. The construction processes of small section prefabrication and large section assembly are adopted. The longest assembled section is 152.6 m, and the maximum lifting weight is about 3200 tons. The main girder is a three-span continuous steel box girder structure with variable sections, and the top plate is an orthotropic steel deck structure with a width of 33.1 m. The beam height of each 5 m section on both sides of the middle pier top is 6.5 m; the beam height of each 37.5 m section on both sides of the pier top changes linearly from 6.5 m to 4.5 m; and the beam height of other sections is 4.5 m. The elevation and cross-section dimensions of the pipeline bridge crossing the Ya13-1 gas field are shown in Figure 1.

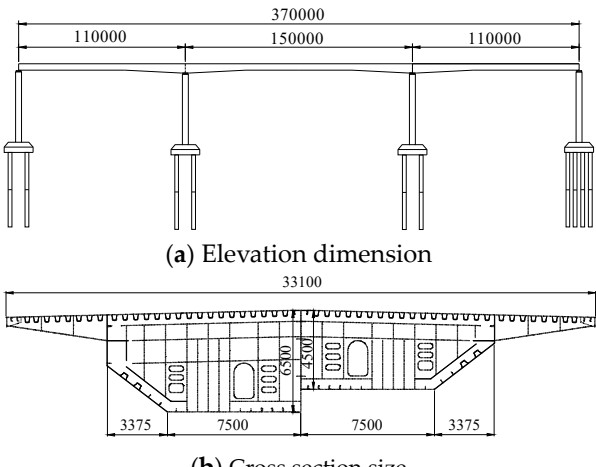

(**a**) Elevation dimension

(**b**) Cross section size

**Figure 1.** Elevation and Cross Section Dimensions of Pipeline Bridge Crossing Ya 13-1 Gas Field (Unit: mm).

### 2.2. Meteorological Conditions

The pipeline bridge across the Ya 13-1 Gas Field is located in Guangdong, Hong Kong, Macao Dawan District, with the geographic coordinates of 22.27° N and 113.82° E. The bridge site has a subtropical marine monsoon climate with an obvious alternation of winter and summer monsoons. In the summer, it is mostly affected by typhoons and is prone to rain and thunderstorms. In winter, there are cold waves and strong winds that cool the temperatures. The highest monthly temperature in summer is 31.4 °C, and the lowest monthly temperature in winter is 14.5 °C. The annual temperature is high, and the sunshine radiation is strong. The monthly temperature change at the bridge site of the pipeline crossing Ya13-1 Gas Field in the past 20 years is shown in Figure 2.

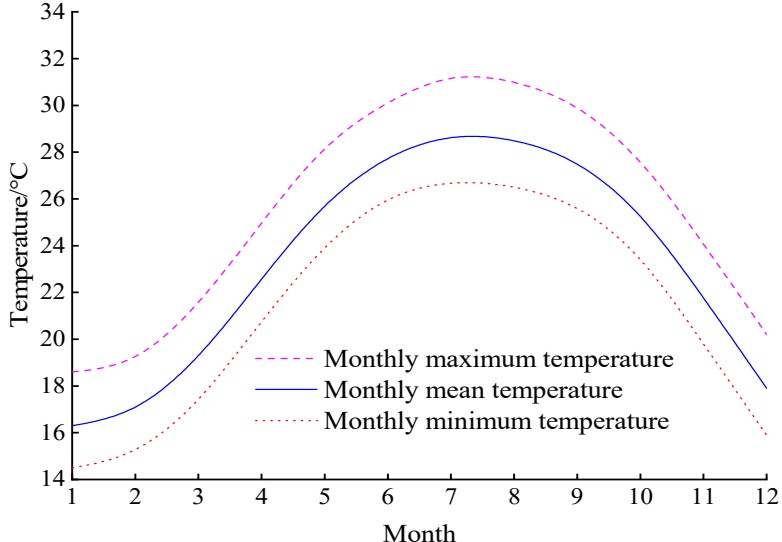

**Figure 2.** Monthly Temperature at Bridge Site in Recent 20 Years (1994~2014).

### 2.3. Closure Process

The pipeline bridge across the Ya 13-1 Gas Field adopts temporary corbel during the closure process, and the specific process is as follows: temporary corbel is set on the roof at the interface of the segment to be closed, and temporary supports are placed on the roof at the interface of the segment that is already in place. During the hoisting process of the closure segment, the corbels are overlapped on the temporary supports, which support and accurately adjust the ends of the closure segment. The setting of temporary corbels and supports is shown in Figure 3.

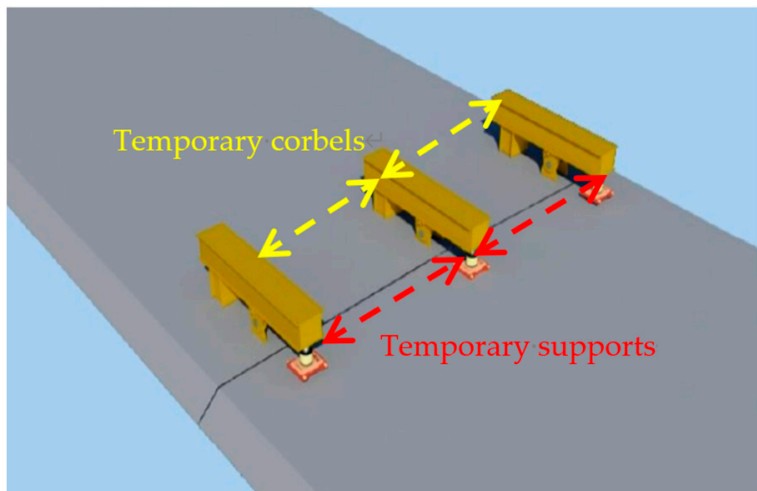

**Figure 3.** Schematic Diagram of Temporary Corbels and Supports for Closure Segment Construction.

After the closure segment is hoisted and initially positioned, the surveyors initially measure the alignment amount in the daytime and then prepare the section steel and steel backing plate according to the measurement data to prepare for the accurate alignment of the beam section at night. The precise positioning of the beam section at night shall be carried out step by step in the order of adjusting the elevation first, then the axis, and finally the mileage. During the appropriate temperature period, the positioning horse plate shall be welded, and the plates shall be connected according to the welding sequence of the top plate, the web plate, and the bottom plate. Finally, the high-strength bolt connection of the U-rib of the inner top plate of the box girder and the butt welding of the U-rib of the outer top plate of the box girder shall be completed.

## 3. Analysis Theory of Transient Temperature Field

### 3.1. Unified Heat Flux Partial Differential Equation

The temperature field distribution of the sea crossing bridge under the action of solar radiation and atmospheric convection meets the applicable conditions of the heat conduction differential equation. Due to the shorter duration of the bridge construction process compared to the operation process, the total heat flux introduced into the bridge during the construction process is much greater than that generated by the internal heat source of solar radiation, so the influence of the internal heat source is not considered. Therefore, according to the Fourier heat conduction theory and in combination with the characteristics of the longitudinal uniform distribution of the bridge temperature field, the heat conduction differential equation of the sea crossing bridge can be derived as follows:

$$\rho c \frac{\partial T}{\partial t} = k \left( \frac{\partial^2 T}{\partial x^2} + \frac{\partial^2 T}{\partial y^2} \right) \tag{1}$$

$$T_{t=0} = T_0(x, y) \tag{2}$$

where:

- $T$ is the temperature at any point in the cross section of the steel box girder;
- $(x, y)$ is the Cartesian coordinate system of the bridge cross section, which are indicated in Figure 4;
- $t$ is the time;
- $\rho$ is the material density;
- $c$ is the specific heat capacity;
- $k$ is the thermal conductivity;
- $T_0(x, y)$ is the initial temperature (minimum temperature) during the construction of the bridge structure.

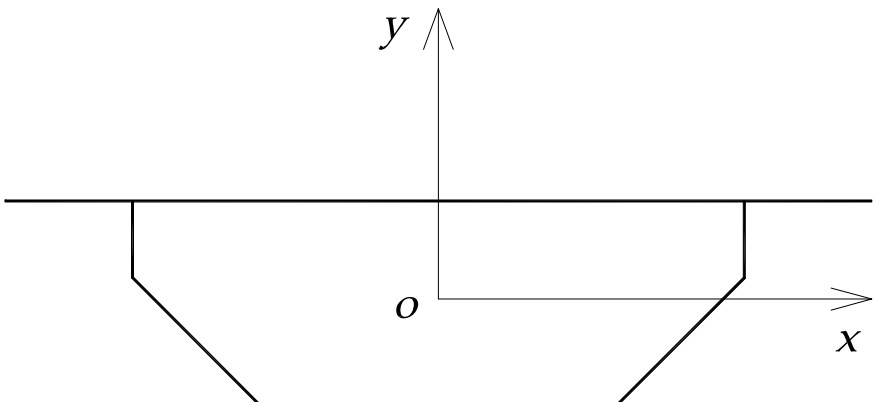

**Figure 4.** Schematic Diagram of Cartesian Coordinate System of Bridge Cross Section.

*3.2. Solution of Partial Differential Equations*

3.2.1. Initial Temperature

According to the monthly temperature statistics of the bridge site over the past 20 years, as shown in Figure 2, and in combination with the construction schedule, it can be determined that the minimum temperature of the bridge closure construction month of the pipeline crossing the Ya13-1 Gas Field is 26 °C, which is set as the initial temperature for solving the heat conduction partial differential equation.

3.2.2. Heat Conduction Boundary Conditions

The steel box girder of the pipeline bridge across the Ya13-1 Gas Field mainly conducts heat exchange with the external environment through solar radiation and atmospheric convection. The heat flux absorbed by solar radiation meets the second boundary condition of heat conduction during the temperature rise of the steel box girder. Then, the heat flux absorbed by convection with the surrounding air meets the third boundary condition of heat conduction. Since the above two heat exchange methods are conducted independently, the second boundary condition of heat conduction can be combined with the third boundary condition, namely:

$$-k\frac{\partial T}{\partial n}\mid_\Gamma = q + h(T_e - T_a) \tag{3}$$

where:

- $q$ is the solar radiation heat flow density, the calculation formula is $q = \beta I$;
- $\beta$ is the solar radiation absorption coefficient of steel;
- $I$ is the solar radiation intensity, and its value is equal to the sum of the solar direct radiation intensity and scattering intensity;
- $h$ is the comprehensive heat transfer coefficient;
- $T_e$ is the external surface temperature of the steel box girder;
- $T_a$ is the air temperature around the steel box girder.

The right end of Equation (3) can be transformed into:

$$q + h(T_e - T_a) = h\left[(T_e + \frac{q}{h}) - T_a\right] = h(T_c - T_a) \tag{4}$$

Formula (3) can be rewritten as follows:

$$-k\frac{\partial T}{\partial n}\mid_\Gamma = h(T_c - T_a) \tag{5}$$

where:

- $T_c$ is the equivalent comprehensive temperature of the external surface of the steel box girder, and the calculation formula is $T_c = T_e + \frac{\beta I}{h}$.

3.2.3. Solar Radiation Intensity

The solar radiation intensity is the sum of direct sunlight intensity and scattering intensity, and the calculation formula is:

$$I = I_1 + I_2 \tag{6}$$

where:

- $I_1$ is the direct sunlight intensity;
- $I_2$ is the sunlight scattering intensity.

The above two parameters can be solved according to the following formula:

$$I_1 = I_d \cos\theta \tag{7}$$

$$I_{\mathrm{d}} = I_0 P_1{}^m \tag{8}$$

$$I_0 = 1353 + 44 \sin\left(\frac{360(89 + D)}{365}\right) \mathrm{W/m^2} \tag{9}$$

$$\cos\theta = \cos\alpha\,\sin h_{\mathrm{s}} + \cos\,\alpha\,\sin h_{\mathrm{s}} \cos(\psi - \psi_0) \tag{10}$$

$$\sin h_{\mathrm{s}} = \sin\phi\sin\delta + \cos\delta\cos\phi\cos\omega \tag{11}$$

$$\sin\psi = \frac{\cos\delta\sin\omega}{\cos h_{\mathrm{s}}} \tag{12}$$

$$\delta = 23.45 \sin\left(\frac{360(284 + D)}{365.25}\right) \tag{13}$$

$$m = 1/\sin h_{\mathrm{s}} \tag{14}$$

$$I_2 = I_0(0.2710 - 0.2913 P_1{}^m)\sin h_s \tag{15}$$

where:

- $I_{\mathrm{d}}$ is the normal solar radiation intensity;
- $\theta$ is the angle of incidence of solar rays;
- $I_0$ is the solar constant;
- $P_1$ is the atmospheric transparency coefficient;
- $h_{\mathrm{s}}$ is the solar altitude angle;
- $D$ is the date serial number;
- $\phi$ is the latitude at the bridge site;
- $\delta$ is the declination of the sun;
- $\omega$ is the solar hour angle;
- $\psi_0$ is the azimuth of the inclined plane.

Formulas (7)–(15) can be used to calculate the solar radiation intensity of the pipeline bridge crossing the Ya13-1 Gas Field from 6:00 a.m. to 18:00 p.m. on the expected closure day. The data for calculating solar radiation intensity in Zhuhai are obtained from [23]. Due to space limitations, the calculation process is omitted. Table 1 gives the specific calculation results of solar radiation intensity.

**Table 1.** Solar radiation intensity at different times (unit: $\mathrm{W/m^2}$).

| Time | 06:00 | 07:00 | 08:00 | 09:00 | 10:00 | 11:00 | 12:00 |
|------|-------|-------|-------|-------|-------|-------|-------|
| T | 26.4 | 182.1 | 284.1 | 397.3 | 476.8 | 537.5 | 569.2 |
| W/B | 0 | 23.9 | 40.2 | 57.3 | 81.2 | 106.3 | 113.2 |
| Time | 13:00 | 14:00 | | 15:00 | 16:00 | 17:00 | 18:00 |
| T | 521.6 | 435.8 | | 328.6 | 215.8 | 126.8 | 14.5 |
| W/B | 104.1 | 78.2 | | 46.4 | 27.6 | 15.7 | 0 |

T value in Table 1 represents the solar radiant intensity of top plate of steel box girder at any time; and W/B value represents the solar radiant intensity of web/bottom plates of steel box girder at any time. The results can be calculated according to Formulas (7)–(15).

### 3.2.4. Equivalent Comprehensive Temperature

According to the calculation formula for the equivalent comprehensive temperature, the calculation of the equivalent comprehensive temperature is only related to the comprehensive heat transfer coefficient after the sunlight radiation intensity is solved. The comprehensive heat transfer coefficient can be expressed as the sum of the solar radiation

heat transfer coefficient and the atmospheric convection heat transfer coefficient. The calculation formula is:

$$h = h_c + h_r \qquad (16)$$

where:

- $h_c$ is the atmospheric convection heat transfer coefficient, the calculation formula is $h_c = 5.8 + 4.0v$;
- $v$ is the wind speed at any time [24];
- $h_r$ is the solar radiation heat transfer coefficient, and the calculation formula is: $h_r = c_s \varepsilon \left[ (T^* + T_c)^4 - (T^* + T)^4 \right]$;
- $c_s$ and $T^*$ are constants, where $T$ is the absolute temperature (Kelvin scale);
- $\varepsilon$ is the radiance.

Since the radiation heat transfer coefficient $h_r$ is related to $T_c$ and $T$, $T_c$ cannot be calculated directly by solving $h_r$, so $T_c$ needs to be calculated iteratively. The MATLAB calculation program is used to solve $T_c$ iteratively. The specific calculation process is as follows: assuming that the initial value $h_{ri}$ at the time $t_i$ is 0, the equivalent comprehensive temperature $T_c(j)$ is calculated by introducing Formulas (5) and (6). At the time $T_c(j+1) - T_c(j) \leq 0.05$, the calculation is considered convergent, and the specific value of the equivalent comprehensive temperature at any moment in the closure stage is obtained. Taking 14:00 as an example, the iterative solution process of equivalent comprehensive temperature corresponding to this time is shown in Figure 5, and the calculation results are shown in Table 2.

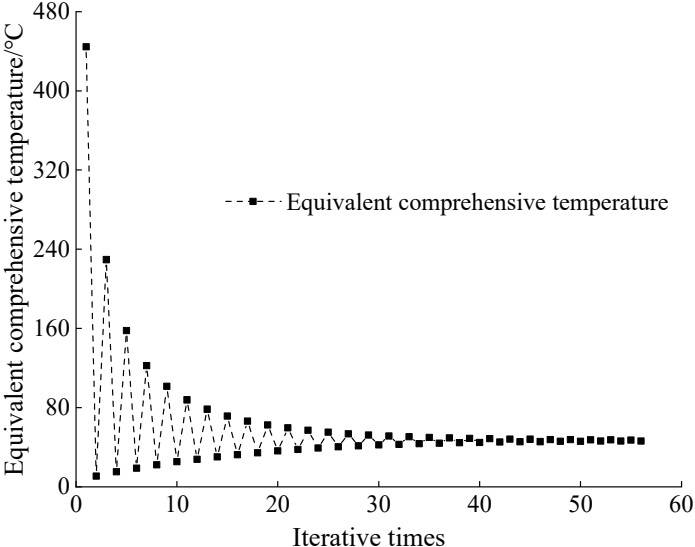

**Figure 5.** Schematic Diagram of Iterative Solution Process of Equivalent Comprehensive Temperature.

**Table 2.** Equivalent Comprehensive Temperature of External Surface of Steel Box Girder (Unit: °C).

| Time | 06:00 | 07:00 | 08:00 | 09:00 | 10:00 | 11:00 | 12:00 |
|------|-------|-------|-------|-------|-------|-------|-------|
| $T_c$ | 28.5 | 31.3 | 32.4 | 33.5 | 35.2 | 37.6 | 39.4 |
| Time | 13:00 | 14:00 | 15:00 | 16:00 | 17:00 | 18:00 | 19:00 |
| $T_c$ | 40.3 | 43.7 | 42.1 | 41.5 | 40.3 | 38.4 | 36.5 |
| Time | 20:00 | | 21:00 | 22:00 | | 23:00 | 24:00 |
| $T_c$ | 33.8 | | 32.5 | 31.4 | | 30.7 | 29.6 |

3.2.5. Fitting of Atmospheric Temperature

The atmospheric temperature around the steel box girder is calculated using the piecewise sine function fitting method [25]. Assuming that the time interval between the daily maximum temperature and the minimum temperature at the bridge site is 12 h, and the atmospheric temperature changes according to the sine law, the fitting calculation formula of the atmospheric temperature change in a day is:

$$T_a = b + k \sin \frac{(h + 30)\pi}{24} \ (0 \le h \le 6) \tag{17}$$

$$T_a = b + k \sin \frac{(h - 10)\pi}{16} \ (6 \le h \le 14) \tag{18}$$

$$T_a = b + k \sin \frac{(3h - 22)\pi}{40} \ (14 \le h \le 24) \tag{19}$$

where:

- $T_a$ is the atmospheric temperature around the steel box girder;
- $b$ refers to the daily average temperature, and its value is $(T_{max} + T_{min})/2$;
- $k$ refers to the daily temperature change amplitude, and its value is $(T_{max} - T_{min})/2$;
- $T_{max}$ refers to the daily maximum temperature;
- $T_{min}$ refers to daily minimum temperature.

According to Formulas (17)–(19) and the temperature statistics at the bridge site in Figure 2, the atmospheric temperature at any time during the closure process can be calculated, as shown in Table 3.

**Table 3.** Atmospheric Temperature at Any Time in Closure Process (Unit: °C).

| Time | 00:00 | 01:00 | 02:00 | 03:00 | 04:00 | 05:00 | 06:00 |
|------|-------|-------|-------|-------|-------|-------|-------|
| $T_a$ | 28.1 | 27.6 | 27.2 | 26.8 | 26.6 | 26.5 | 26.4 |
| Time | 07:00 | 08:00 | 09:00 | 10:00 | 11:00 | 12:00 | 13:00 |
| $T_a$ | 28.9 | 29.9 | 31.0 | 32.1 | 33.2 | 34.3 | 35.3 |
| Time | 14:00 | 15:00 | 16:00 | 17:00 | 18:00 | 19:00 | 20:00 |
| $T_a$ | 37.8 | 37.6 | 37.2 | 36.4 | 35.5 | 34.3 | 33.0 |
| Time | 21:00 | | 22:00 | | 23:00 | | 24:00 |
| $T_a$ | 31.7 | | 30.3 | | 29.1 | | 28.1 |

## 4. Numerical Analysis of Transient Temperature Effect

### 4.1. Transient Temperature Gradient Analysis

Based on the analysis results of the above parameters and boundary conditions, the large general finite element program ABAQUS is used to predict and analyze the temperature gradient at any time during the closure of the main girder of the pipeline bridge across the Ya13-1 Gas Field. The three-dimensional heat conduction unit DS4 is used to establish the solid finite element model of the steel box girder, and the three-dimensional heat conduction unit DC3D8 is used to establish the air model around the steel box girder. The longitudinal unit size of the steel box girder is 1 m, and the cross section is established according to the actual size. The above equivalent comprehensive temperature and atmospheric temperature are input into the finite element model as the temperature field boundary of the steel box girder for thermal conduction numerical analysis. The transient temperature field analysis adopts the Heat Transfer (Transfer) analysis step. The calculation starts at 6:00 a.m. and ends at 24:00 a.m. The finite element solid model for the analysis of the transient temperature field of the steel box girder of the pipeline bridge

across the Ya13-1 Gas Field during the closure phase is shown in Figure 6, and the material parameters used for modeling are shown in Tables 4 and 5.

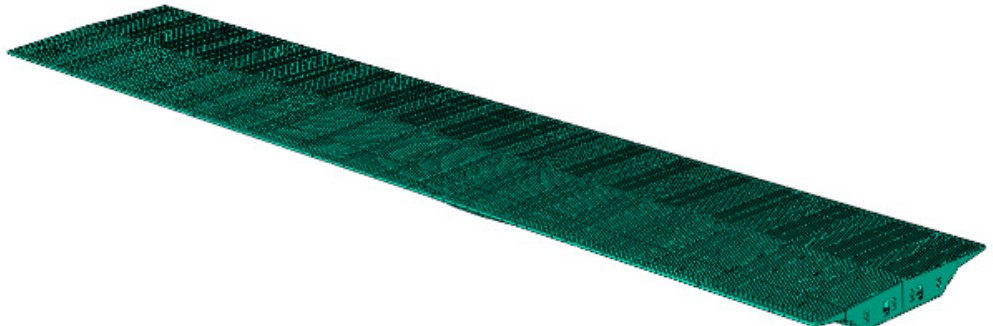

**Figure 6.** Finite element solid model for transient temperature field analysis.

**Table 4.** Material Parameters of Steel Box Girder and Surrounding Air.

| Parameters | Density (kg·m$^{-3}$) | Specific Heat Capacity [J·(kg·°C)$^{-1}$] | Thermal Conductivity [W·(m·°C)$^{-1}$] | Absorptivity | Irradiance |
|---|---|---|---|---|---|
| air | 1.293 | 1.000 | 1.000 | — | — |
| steel | 7854 | 434 | 60 | 0.55 | 0.60 |
| Parameters | Elastic modulus (GPa) | Poisson's modulus | | | |
| air | — | — | | | |
| steel | 2.06 | 0.25 | | | |

**Table 5.** Mechanical Property Indexes of Steel.

| Parameters | Yield Strength (MPa) | Tensile Strength (MPa) | Elongation (%) |
|---|---|---|---|
| steel | 345 | 490 | 20 |

The vertical temperature gradient effect of the section of the pipeline bridge across the Ya13-1 gas field has a significant impact on the closure process. Therefore, the vertical temperature gradient of the steel box girder needs to be taken as an important parameter to control the temperature effect during the closure process. Through the finite element analysis of the transient temperature field, the temperature difference (predicted value) between the top and bottom plates at any time during the closure of the steel box girder can be calculated. At the same time, in order to verify the correctness of the prediction model of the transient temperature field, the real-time monitoring of the temperature field was carried out during the installation of the large section of the side span. The comparison results between the predicted value and the measured value of the temperature difference between the top and bottom plates at any time during the closure process are shown in Figure 7. The vertical distribution of temperature along the steel box girder is shown in Figure 8.

It can be seen from Figure 7 that the predicted temperature difference between the top and bottom plates of the steel box girder at each time during the closure process is close to the measured value (the maximum deviation is less than 2 °C), and the predicted value is slightly greater than the measured value, making the structure safer. The predicted temperature difference between the top and bottom plates is consistent with the measured value. From 6:00 a.m. to 24:00 p.m., the temperature difference first increases and then decreases. The maximum temperature difference between the top and bottom plates occurs at about 14:00 p.m. on the day of closure. It can be seen from Figure 8 that the vertical temperature gradient of the steel box girder during closure has obvious nonlinear

characteristics. The temperature gradient is high within 0.4 m from the top of the roof, and tends to zero beyond 0.4 m. Therefore, the calculation of the transient temperature effect of the steel box girder can be conducted using the temperature gradient mode, as is shown in Figure 9. The parameter *H* is the height of steel box girder, and there is a temperature gradient effect within the range of 0~0.4 m from the top plate, and the temperature linearly decreases from T1 to T2. Within the range of 0.4 m from the top plate to the bottom plate, the temperature remains unchanged at T2, and there is no temperature gradient effect within this range.

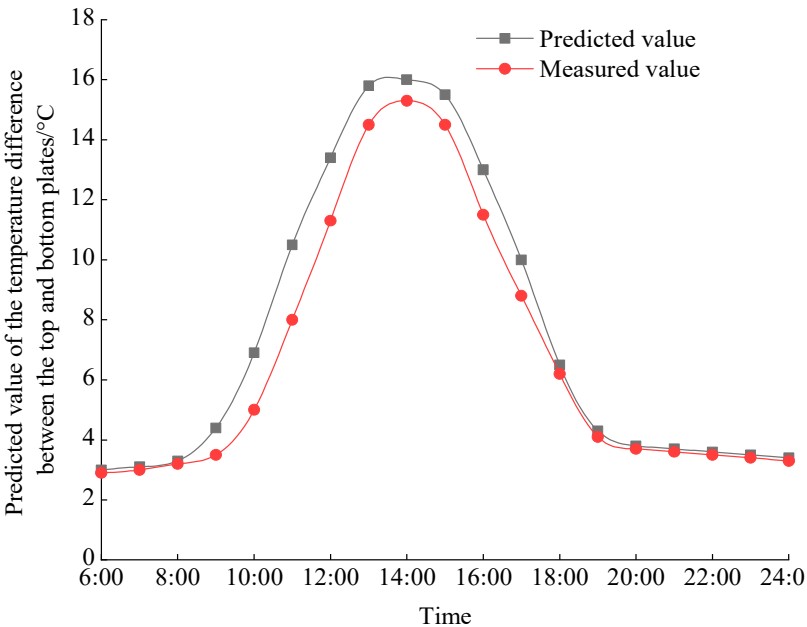

**Figure 7.** Comparison between predicted and measured temperature difference between top and bottom plates.

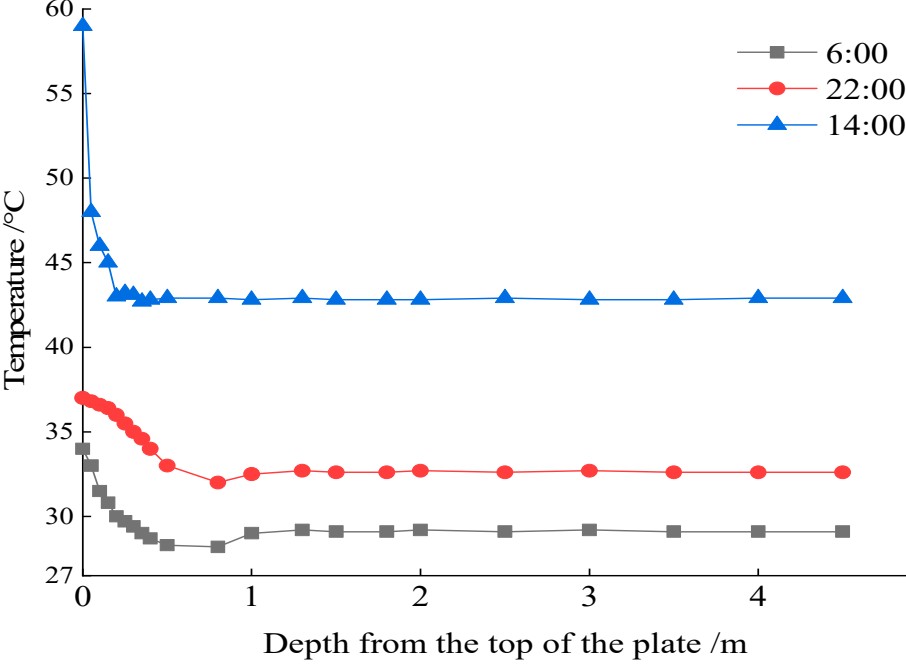

**Figure 8.** Vertical Distribution Characteristics of Temperature along Steel Box Girder.

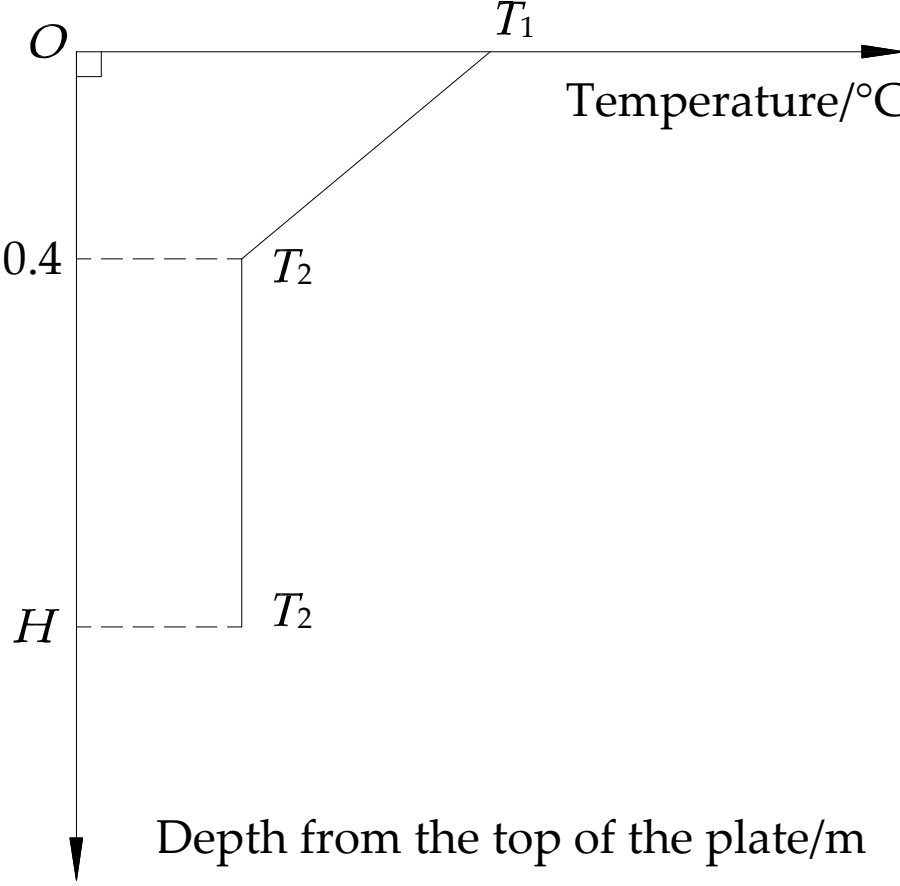

**Figure 9.** Temperature Gradient Model of Steel Box Girder.

*4.2. Refined Analysis of Transient Temperature Effect*

The influence of the transient temperature effect on the structure during the bridge closure of the pipeline crossing the Ya13-1 Gas Field is mainly reflected in the following three aspects: (1) the influence on the smoothness of the large segment joint; (2) the influence on the length of the closure section; and (3) the influence on the coincidence between the finished bridge alignment and the designed alignment. The above three issues are essentially the impact of the temperature effect on structural alignment. Therefore, the following is a detailed analysis of the impact of the transient temperature effect on structural alignment during the closure of the steel box girder. The specific idea is: after the analysis of the transient temperature field of the steel box girder in Section 3.1, apply the transient temperature gradient to the structure to calculate the length change of the closure opening under the effect of the transient temperature gradient during the closure of the steel box girder, the corner of closure beam end, and the finished bridge alignment. The large general finite element program, ABAQUS, is still used to calculate the transient temperature effect. The steel box girder is simulated by the 4-node quadrilateral reduced integral element S4R plate shell element. The boundary conditions are linear and plane constraints. At the fixed support, 5 degrees of freedom of the steel box girder bottom plate node are constrained to release the degrees of freedom of rotation along the bridge direction (as shown in Figure 10). At the movable support, only the vertical degrees of freedom of the steel box girder bottom plate node are constrained. In the construction phase, multiple analysis steps are defined and the model change function is adopted. The Static (General) analysis step is adopted.

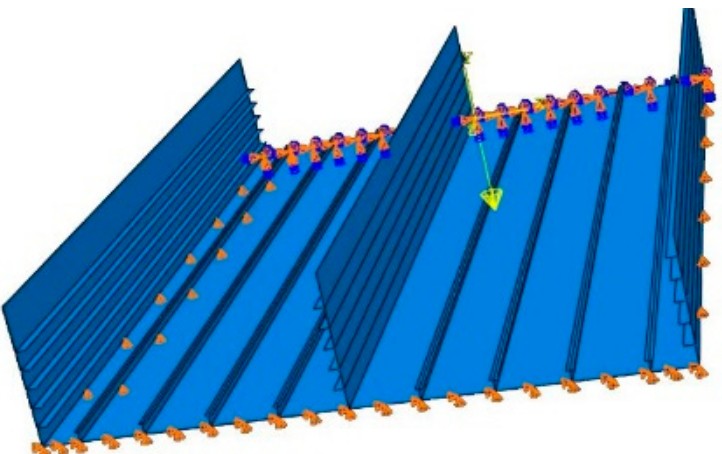

**Figure 10.** Schematic Diagram of Boundary Conditions at Fixed Supports.

### 4.2.1. Closure Length and Beam End Corner

Figure 11 is a histogram of the axial expansion and contraction of the closure opening of the steel box girder and the change in the beam end angle caused by the transient temperature gradient at any time during the closure process. It can be seen from the figure that under the action of the transient temperature gradient, the axial expansion and contraction of the closure opening of the steel box girder are always positive, which means that the length of the closure section is shorter than the design and reaches the maximum at 14:00 p.m., with a value of 26.7 mm. The angle of the cantilever end section also increases with the increase in the transient temperature gradient, and reaches the maximum value at 14:00 p.m., which is 1.274°. From 21:00 in the evening to 6:00 in the morning of the next day, the values of the axial expansion of the steel box girder closure opening and the angle of the closure opening beam end are small, and the change amplitude tends to be gentle.

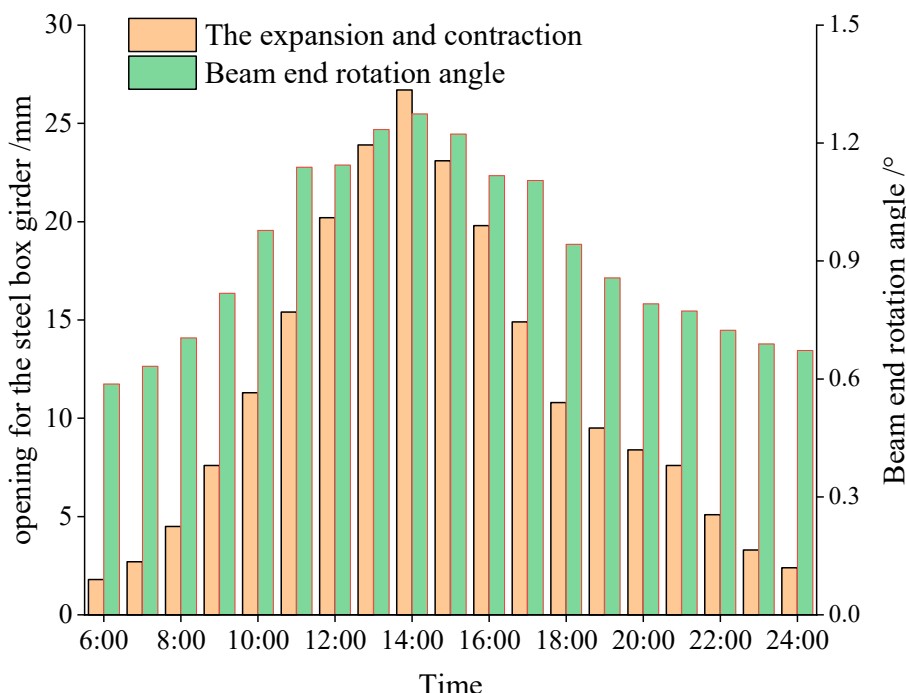

**Figure 11.** Schematic Diagram of Axial Expansion and Beam End Corner.

### 4.2.2. Bridge Alignment after Closure

Figure 12 shows the change curve of the bridge forming the alignment of the structure caused by the transient temperature gradient at any time during the closure process. It can

be seen from the figure that the temperature gradient has a greater impact on the bridge forming the alignment of the structure. The greater the temperature gradient, the greater the deviation of the corresponding bridge forming the alignment. When the closure is carried out at the time when the temperature gradient is maximum (14:00 p.m. on the day of closure), and the section temperature after closure tends to be the same and the bridge forming the alignment error of the side span and the middle span reaches the maximum at the same time, 51.1 mm and 118.3 mm, respectively. Therefore, effective measures should be taken to control the influence of temperature difference on structural alignment after closure.

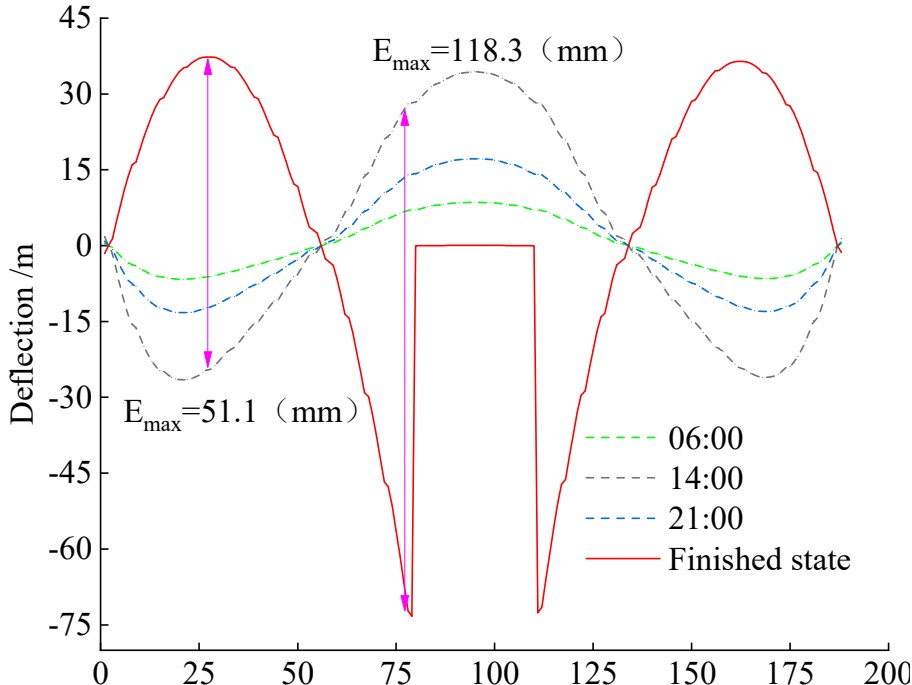

**Figure 12.** Alignment Deviation of Completed Bridge.

*4.3. Discussion on Optimal Closure Period and Beam End Angle Control*

4.3.1. The Best Closing Time

The following three factors should be considered to determine the optimal closure time under the effect of transient temperature gradient during the closure of the pipeline bridge across the Ya13-1 Gas Field:

1. The length change in the closure section caused by the transient temperature effect is the smallest during the closure process;
2. During closure, the angle of the closure beam caused by the transient temperature effect is the smallest;
3. During the closure process, the alignment deviation between the closed bridge and the completed bridge caused by the transient temperature effect is the smallest.

According to the provisions on the minimum weld width and weld seam reserve of the pipeline bridge across the Ya13-1 Gas Field, the maximum adjustment of the weld seam length error to the top and bottom plates of the steel box girder is 10.4 mm. At the same time, according to the calculation of the cutting amount of the top and bottom plates of the steel box girder caused by the angle of the closure beam end, when the angle of the closure beam end is below 0.8°, the maximum value of the cutting amount of the top and bottom plates is less than 3 mm, so the cutting process of the top and bottom plates can be omitted to simplify the construction process. The alignment error of the pipeline bridge across the Ya13-1 Gas Field is required to be 10 mm. According to Figure 7, the corresponding vertical temperature difference between the top and bottom is 3 °C. Based on the above

analysis, according to the three control conditions that the length change of the closure section is less than 10 mm, the angle of the closure mouth beam end is less than 0.8°, and the vertical temperature gradient of the top and bottom plates is less than 3 °C. Therefore, it can be concluded from Figures 6 and 7 that the optimal closure period of the pipeline bridge across the Ya13-1 Gas Field is from 21:00 p.m. of the closure day to 6:00 a.m. the next day.

Closure construction shall be carried out according to the above optimal closure period. During the closure process, through real-time monitoring of the temperature sensors at the top and bottom plates of the steel box girder, it is concluded that the vertical temperature gradient of the top and bottom plates during the optimal closure period is within 2 °C. At the same time, through the real-time monitoring of the closure opening length and the beam end angle, it is concluded that the length of the closure section changes within 6 mm during the optimal closure period, and the closure opening beam end angle is less than 0.4°. From the above results, it can be seen that the predicted results of the temperature field of the steel box girder of the pipeline bridge across the Ya13-1 Gas Field are consistent with the measured values and the theoretical analysis results of the transient temperature effect are accurate. The method proposed in this paper can provide a reference for the analysis of the transient temperature effect during the closure of the steel box girder bridge across the sea and guide actual engineering construction.

4.3.2. Discussion on Angle Control of Closure Beam End

During the closure process of the pipeline bridge across the Ya13-1 Gas Field, even if closure construction is carried out in the best closure period, there are small corners at both ends of the closure opening. In order to make the corners at both ends of the closure as small as possible to facilitate the welding closure construction and meet the requirements of smooth alignment after closure, it is recommended to take the following measures to control the corners of the closure beam end during closure.

1.  A temporary positioning jack shall be set near the support of the side span large segment to rotate the beam body by adjusting the elevation of the support of the side span large segment, thus eliminating the corner of the closure beam end. The adjustment diagram is shown in Figure 13.

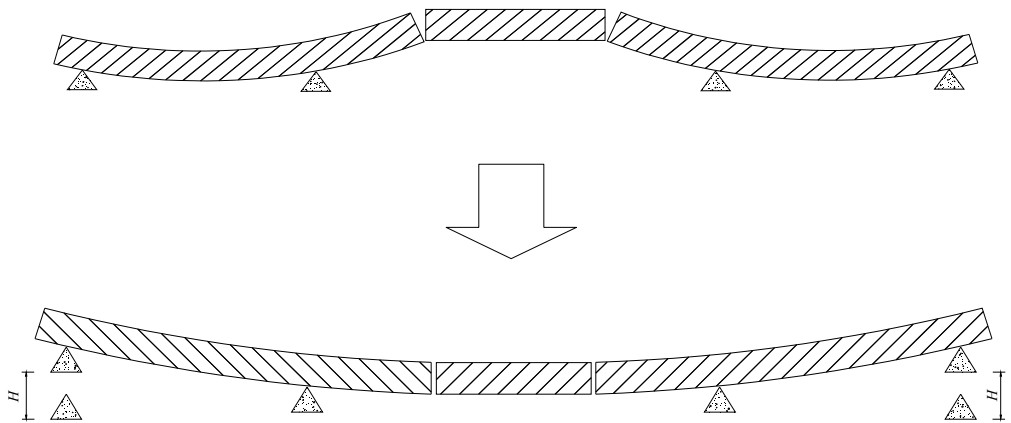

**Figure 13.** Schematic Diagram of Temporary Positioning Jack Adjusting the Corner of Closure Beam End.

2.  A temporary counterweight is applied at the cantilever end of large section of the side span to make the end section of the closure beam produce a reverse rotation angle and then offset the beam end rotation angle caused by the transient temperature effect. The adjustment diagram is shown in Figure 14.

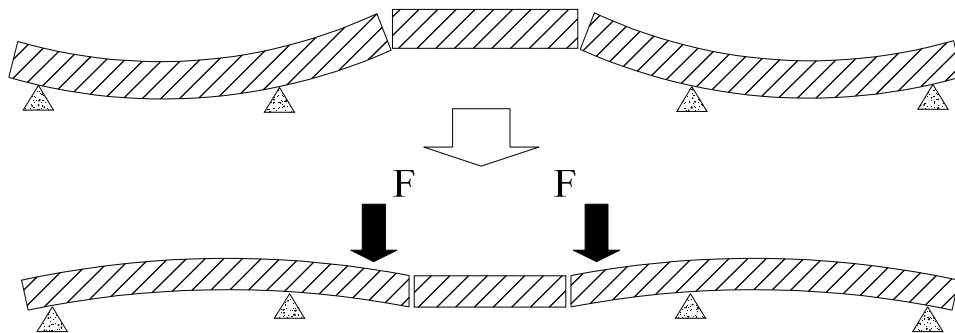

**Figure 14.** Schematic Diagram for Temporary Counterweight Adjustment of Closure Beam End Corner.

**5. Conclusions**

1. The theoretical analysis of transient temperature field combined with the finite element numerical simulation method can effectively predict the transient temperature gradient of the pipeline bridge across the Ya13-1 gas field at any time during the closure process. The results show that the predicted value of the transient temperature gradient is close to the measured value (the maximum deviation is less than 2 °C), and the predicted value is slightly greater than the measured value, making the structure safer. The predicted value of the transient temperature gradient is consistent with the measured value. From 6:00 a.m. to 24:00 p.m., the temperature gradient increases first and then decreases. The maximum temperature difference between the top and bottom plates occurs at about 14:00 p.m. on the day of closure. During closure, the vertical temperature gradient of the steel box girder has obvious nonlinear characteristics. Within 0.4 m from the top of the roof, the temperature gradient is relatively high, while beyond 0.4 m, the temperature gradient gradually tends to be towards zero.

2. According to the three control conditions that the length change of the closure section is less than 10 mm, the angle of the closure beam end is less than 0.8°, and the vertical temperature gradient of the top and bottom plates is less than 3 °C, it is concluded that the best closure period of the pipeline bridge across the Ya13-1 gas field is from 21:00 p.m. of the day of closure to 6:00 a.m. the next day. The problem of small corners at both ends of the closure opening during the optimal closure period can be eliminated by setting temporary adjusting jacks or applying temporary counterweights to facilitate the welding closure construction and meet the requirements of smooth alignment after closure.

3. The construction of cross-sea bridges in offshore waters is affected by strong solar radiation and atmospheric convection during the construction process. The temperature gradient inside the structure has a large scope of action, and the value of the temperature gradient is significantly different at different times of the day. Therefore, for the selection of the temperature gradient calculation model in the construction process of the sea crossing bridge, the unified model specified in the specifications cannot be simply applied, but the real temperature gradient model at any time inside the structure should be obtained through the transient temperature field analysis theory combined with the finite element numerical simulation in combination with the monitoring data of the temperature field at the bridge site over the years. This will help to achieve a refined solution and prediction of the transient temperature effect of the structure and then guide the actual engineering construction.

**Author Contributions:** Conceptualization, Z.L. and Y.L.; methodology, Z.L. and Y.L.; software, Z.L. and F.D.; validation, Z.L.; investigation, Z.L. and Y.L.; data curation, Y.L. and F.D.; writing—original draft preparation, Z.L. and J.W.; writing—review and editing, J.W. and F.D.; supervision, F.D. All authors have read and agreed to the published version of the manuscript.

**Funding:** This research was funded by the Fundamental Research Funds for the Central Universities, CHD (300102213522), the General Program of Shanxi Province Natural Science Foundation (202203021221025), the Natural Science Foundation of Jiangsu Province (Grant No. BK20200793), and the Science and Technology Innovation Program of Shanxi Province Higher Education Institutions (2021L010).

**Institutional Review Board Statement:** Not applicable.

**Informed Consent Statement:** Not applicable.

**Data Availability Statement:** The raw/processed data required to reproduce these findings cannot be shared at this time as the data also forms part of an ongoing study.

**Acknowledgments:** The author thanks all the researchers involved in theoretical analysis, numerical simulation, and data collection and validation. They are members and students of the research team in the field of bridge engineering in Shanxi University, Nanjing Forestry University and Chang'an University.

**Conflicts of Interest:** The authors declare no conflict of interest.

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
