# Peer review of "Refined Analysis of the Transient Temperature Effect during the Closing Process of a Cross-Sea Bridge"

_sustainability, doi:10.3390/su151712970_

Round 1
Reviewer 1 Report
Based on a careful analysis, I can formulate the following remarks:
1) The aim of this article, based on the authors’ scrupulous investigations, is to propose an efficient study of the transient temperature effect during the construction of the cross-sea bridge off the coast, by means of a refined analysis and prediction of transient temperature gradient.
2) The topic represents in my opinion a relevant approach of the proposed theme in the field, based on meticulous theoretical investigations, correlated with simulation results.
3) In comparison with other published material, the authors' contribution adds to the subject area a new approach/methodology, with several significant contributions.
They performed a scrupulous a theoretical analysis, validated by meticulous simulation results.
They conducted a structural response under the conditions of strong solar radiation and atmospheric convection using the method of combining theoretical research and numerical simulation.
Firstly, the partial differential equation of uniform heat flux density on the outer surface of the main girder under the action of solar radiation and atmospheric convection was established. They obtained the equation by calculating the solar radiation intensity and the comprehensive heat transfer coefficient, as well as fitting the atmospheric temperature on the outer surface of the main girder, and finally results the equivalent comprehensive temperature at any time of the main girder.
Secondly, the authors established a numerical analysis model of the heat conduction of the main girder section, by imputing the equivalent comprehensive temperature into the numerical model as the temperature field boundary to solve the transient temperature gradient of the section. By comparison with the measured data, they verified the so-obtained values.
Finally, they applied the transient temperature gradient to the girder, and the temperature effect of the main girder during the closing process was calculated.
They also discussed the construction control measures. The research results show that the predicted value of the transient temperature gradient is consistent with the measured value (the maximum deviation is less than 2 ℃), and the predicted value is slightly larger than the measured value, which makes the structure safer.
They established that during the closing process, the temperature gradient of the main girder has obvious non-linear characteristics; the best closing time of the main girder is from 21:00 in the evening of the closing day to 6:00 AM of the next day.
For the small angles at both ends of the closure segment during the best closing time, they adopted a temporary adjustment jacks and temporary counterweights in order to eliminate the small angles at both ends of closure segment in order to facilitate the welding construction and meet the smoothness requirements of bridge alignment.
The research results have practical significance for improving the actual calculi and predictions.
4) It is well-known fact that during the construction of sea-crossing bridges, due to the influence of solar radiation, atmospheric convection, annual temperature change, sudden cooling and other factors, uneven temperature fields are generated inside the structure, and the resulting temperature stress and deformation will have adverse effects on the structure.
The temperature effect of bridge structures generally can be annual temperature change, sunshine temperature, respectively sudden cooling.
For cross-sea bridges, exposed to the marine environment for a long time, the sunshine temperature effect is particularly significant. Under extreme conditions, the temperature effect caused by strong sunshine radiation and atmospheric convection even exceeds the dead load and live load effects and becomes the first control factor in bridge design and construction, it seriously threatens the smooth construction and normal operation and maintenance of cross-sea bridges.
Many scholars have carried out a lot of research on the temperature effect of bridge structures, mainly using the measured data of temperature field combined with FEM simulation to study the temperature field distribution and temperature effect of bridges of various materials and structural forms under the external meteorological conditions, and obtained some valuable conclusions.
The existing research mainly focuses on the temperature field distribution mode of inland bridges under conventional meteorological conditions.
However, there are not enough scientific works and analyzes focus on temperature field distribution and transient temperature effect of offshore bridges under strong solar radiation and atmospheric convection environment.
Additionally, most of the existing studies, based on the measured temperature data, cannot accurately predict the temperature effect of the sea-crossing bridge at any time in the absence of on-site measured data.
For continuous steel box girder bridges constructed by large segment assembly method, the temperature effect will have a great impact on the length of closure section, the smoothness of closure joint and the finished bridge alignment.
Predicting the temperature effect of the structure at any time in the closure process in advance, and selecting the appropriate closure time to ensure the smooth closure process is a technical problem faced in the current construction process of sea-crossing bridges.
Therefore, the authors, based on the lack of measured temperature data, propose a new strategy, how to carry out refined analysis and prediction of the transient temperature gradient model of cross-sea bridges under complex marine meteorological conditions. Such complex conditions are the strong sunlight radiation and atmospheric convection. In these cases, they have to offer the entire distribution characteristics of the overall transient temperature field of the structure, and to achieve accurate calculation of the transient temperature effect during the key construction stages of cross-sea bridges.
These issues represent significant subjects in order to propose corresponding construction control measures to ensure the smooth construction of cross-sea bridges, which needs further study.
The authors, based on a given bridge, address the issue of refined analysis of temperature field distribution and transient temperature effect of offshore bridges under strong solar radiation and atmospheric convection environment, using the method of combining theoretical research and numerical simulation.
Their new approach also allows for a more efficient and effective solution to the complex problem of predicting the temperature effect of the sea-crossing bridge at any time in the absence of on-site measured data.
Additionally, the proposed method provides important guidance for the optimal timing of closure and the selection of construction control measures during the closure process.
Overall, this study is important as it offers a new approach to accurately analyze and predict the temperature gradient effect during the closure process of cross-sea bridge, propose reasonable construction control measures, and ensure the smooth construction of the cross sea bridge.
Some of their main contributions are:
· the theoretical analysis of transient temperature field combined with the FEM analysis can effectively predict the transient temperature gradient of the pipeline bridge at any time during the closure process;
· they can predict the optimal closure time period;
· the strong solar radiation and atmospheric convection during the construction process have significant influence;
· therefore, for the selection of the temperature gradient calculation model in the construction process of the sea-crossing bridge, the unified model specified in the specifications cannot be simply applied;
· the real temperature gradient model at any time inside the structure are obtained through the transient temperature field analysis theory combined with the FEM simulation in combination with the monitoring data of the temperature field at the bridge site over the years, so as to achieve the refined solution and prediction of the transient temperature effect of the structure, and then guide the actual engineering construction.
5) In my opinion, the presented conclusions are suitable related to their research results and prove that they reached the proposed goal.
6) The references in my opinion are very appropriate and their number underlines the scrupulosity of the authors.
7) In this paper, the graphical illustration is well conceived and realized and consequently they contribute to a better understanding of the performed theoretical investigations as well as to underlining the usefulness of the simulation results for the described approach.
The obtained performances are very promising and I express my hope to continue their very interesting and useful research. I encourage publishing in a new contribution their further results.
Author Response
We would like to thank the Reviewer for reviewing our manuscript and providing insightful comments, which have helped improve its overall quality.
Reviewer 2 Report
1. The authors provide drawings that are not very informative, for example figure 3. This figure does not contain explanatory inscriptions except for the name of the figure. Therefore, it is extremely difficult for the reader to understand the essence of the problem, which must be dealt with during construction. Authors should explain in detail to readers why these calculations are needed.
2. The authors present differential equations containing x, y axes. But these axes are not indicated in any way in the figures. The authors do not explain why formula (1) takes into account only thermal conductivity through the Laplace operator, but does not take into account internal heat sources (heating of structures by solar radiation over a considerable length.
3. The equations are difficult to understand, since the same notation has different content. For example, T is the temperature at any point in the cross section of the steel box girder in formula (1);
T is the air temperature around the steel box girder in formulas (3), (17), (18), (19); The authors do not specify what temperature is taken in this case (in Celsius or Kelvin). According to Table 1 Celsius degrees are used, however absolute temperature should be used in formula (16).
4. The authors do not explain the source of data on changes in solar radiation activity over time and air temperature outside the bridge, on the basis of which the data of tables 1, table 2, and table 3 were calculated.
4. At the first stage, the authors do not take into account the cooling of structures due to convective air flows and radiation from the structures themselves. As a result, according to table 1, the temperature of the structures at 13:00 is an incredible 521.6.
5. The authors do not explain what the W/B value shows and how it is calculated. They only indicate that W/B represents web/bottom plate of steel box girde
6. In Table 3, the authors do not explain what Tc is. If this is the temperature of atmospheric air, then it is designated as T.
7. The material properties used in the calculation do not contain the elastic modulus and Poisson's modulus, which are necessary for calculating the elastic stresses and deflections of the structure.
8. The authors give only a schematic figure 8, without indicating the actual values of T1 and T2 for any point in time.
Reviewer 3 Report
The paper “Refined analysis of transient temperature effect during closing process of cross-sea bridge” numerically investigates the transient temperature effects on a pipeline bridge during closure process. The bridge is 370 m long and has a three span continuous steel box girder and an orthotropic steel deck. For transient temperature field analyses, 3D finite element model of the bridge was generated using ABAQUS software. The obtained analysis results showed that the predicted values of the transient temperature gradient were consistent with the measured values. I think this paper is a useful work applicable in engineering practice and I recommend that it be published as is in Sustainability.
Author Response

(The authors gave the same response as above.)

Round 2
Reviewer 2 Report
The authors have eliminated the comments that were made to the first version of the article. The revised version of the manuscript may be published